# Exploring Professional Practice Environments and Organisational Context Factors Affecting Nurses’ Adoption of Evidence-Based Practice: A Scoping Review

**DOI:** 10.3390/healthcare12020245

**Published:** 2024-01-18

**Authors:** Luís Furtado, Fábio Coelho, Natália Mendonça, Hélia Soares, Luís Gomes, Joana Pereira Sousa, Hugo Duarte, Cristina Costeira, Cátia Santos, Beatriz Araújo

**Affiliations:** 1Department of Nursing, Mental Health and Gerontology, School of Health, University of the Azores, 9700-042 Angra do Heroísmo, Portugal; 2Faculty of Health Sciences and Nursing, Universidade Católica Portuguesa, 1649-023 Lisboa, Portugal; 3Flores Island Healthcare Unit, 9960-430 Flores Island, Portugal; 4Center for Innovative Care and Health Technology—ciTechCare, School of Health Sciences, Polytechnic of Leiria, 2411-090 Leiria, Portugal; 5Center for Interdisciplinary Research in Health, Faculty of Health Sciences and Nursing, Universidade Católica Portuguesa, 4169-005 Porto, Portugal

**Keywords:** evidence-based practice, nursing, nursing care, nursing administration research, implementation science, nurse administrators

## Abstract

This scoping review, conducted within the Joanna Briggs Institute (JBI) framework, analysed the recent literature (January 2018 to March 2023) addressing factors inherent to professional practice environments and organisational contexts influencing nurses’ adoption of evidence-based practice (EBP). This review included studies involving nurses regardless of sector, practice setting, and scope of practice. A systematic search was undertaken across the PubMed, Web of Science, CINAHL, and MEDLINE databases, as well as the EThOS, OATD, and RCAAP platforms. The extracted textual elements underwent a content analysis, resulting in a coding structure established through an inductive approach that categorised information into main categories and subcategories linked by similarity and thematic affinity. Forty-one studies were included, revealing four main categories of factors impacting EBP adoption by nurses: (1) organisational dynamics, (2) management and leadership, (3) teamwork and communication, and (4) resources and infrastructure. The study’s limitations acknowledge the subjective nature of categorisation, recognising potential variations based on individual perspectives despite adopting procedures to minimise the risk of bias. The results provide a substantial foundation for developing interventions to cultivate environments conducive to EBP adoption by nurses, thereby enhancing the integration of evidence into nurses’ professional practice contexts. This review was prospectively registered on the Open Science Framework (registration no. osf.io/e86qz).

## 1. Introduction

The concept of evidence-based medicine (EBM), a precursor to the concept of evidence-based practice (EBP), refers to the conscious, explicit, and judicious use of the best evidence in the decision-making process concerning care for a person, considering their values and circumstances [1]. EBP, in turn, is understood as a fundamental resource for professional practice in the health sector which is orientated towards solving problems originating in clinical practice, using the best external evidence and combining it with the preferences and values of the person being cared for, the expertise of a clinical professional, and information from patient data, also known as internal evidence [2].

Healthcare professionals should base their interventions on current and robust scientific evidence to achieve better health outcomes, higher quality of care [3], and increases in the cost-effectiveness of healthcare and to enhance the efficiency and sustainability of healthcare systems [4]. Failure to translate the best available evidence into professional practice translates into inconsistency, variability in care, and suboptimal results [3,5]. And even with its undisputable decisive importance, the literature has shown that EBP is not a widely used approach in the practice of health professionals, which naturally include nurses. This condition represents a considerable challenge for health systems [6], as it is estimated that 30–40% of health service users do not receive care based on the best available evidence [7].

The need to incorporate EBP into clinical practice is well-established and well-founded; however, its application is still hampered by the perceptions that many nurses have about EBP, which translate into barriers to the use of research in their care activity [8], resulting in a worrying situation considering that EBP is related to a better-informed professional decision-making process, as well as a more remarkable ability to plan and provide efficient, individualised, and person-centred nursing care [9].

Successive studies that have focused on how nurses view EBP indicate that this professional group values the approach but suffers from inconsistent implementation subject to various constraints and shortcomings, including the inadequacy of the facilities of healthcare organisations for the development of EBP and a lack of time, resources, institutional support, individual knowledge, and specific competences in EBP, but also autonomy and funding or even limitations in access to sources of evidence [10,11,12], a circumstance that intensifies the gap between theory and practice [13].

The contexts in which care is provided are nonlinear, diverse, dynamic, complex, and adaptive, characterised by networking, with interactions at various levels and in different locations, influenced by multiple values and the different behaviours of the various players but also by organisational limits, external pressure, and environmental factors, which is why it is now assumed worldwide that the organisational context is a determining factor for the implementation and adoption of EBP and thus also for mitigating the gap between theory and practice [14,15].

Implementing EBP, duly supported by the healthcare organisation and its managers (at different hierarchical levels), allows nurses to assume their role as agents of change, facilitating their professional autonomy, with the positive impact it has on the health outcomes of those who receive their care [16,17]. Establishing EBP as a nursing priority is a commitment that must be made in all respects [18]. Therefore, it is necessary to determine which factors currently inhibit the adoption of EBP.

Due to the relevance of carrying out this scoping review of the literature, and as part of the respective preparatory work, a preliminary search was carried out on the PROSPERO, Open Science Framework, Cochrane Database of Systematic Reviews, and Joanna Briggs Institute (JBI) Evidence Synthesis platforms, and no recently published or ongoing literature reviews on this specific topic and purpose were identified. Therefore, considering the multitude of existing studies on the subject, the diversity of studies (primary and secondary) on this topic (which use different methodologies, include diverse populations, and focus on various dimensions of the phenomenon), and the relevance of the subject to the professional and disciplinary area of nursing, with a particular focus on the management of health and nursing services, this study aimed to systematically analyse the findings of the literature published over the last five years on factors associated with professional practice environments and the organisational context with an impact on the adoption of EBP by nurses, to design an intervention plan aimed at promoting environments favourable to the adoption of EBP and, thus, the integration of scientific evidence by nurses in their respective professional practice contexts. Taking into account the heterogeneity of the potential sources of evidence included in this review, and also the fact that the aim was to map and organise factors and thus also areas of intervention, identifying in the process any knowledge gaps that may present opportunities for future research, a scoping review of the literature methodology emerged as the natural and most appropriate alternative response to this design [19].

## 2. Method

### 2.1. Research Question

To accomplish the aim of this literature review study, a review question was defined and structured according to the PCC mnemonic (“population”, “concept”, and “context”) [20,21], opting to omit the “context” component in the formulation of the research question as there was no need to circumscribe this parameter during the search. This study, therefore, sought to answer the following question: what factors related to professional practice contexts and organisational contexts impact the adoption of EBP by nurses?

### 2.2. Study Design

The literature review was conducted throughout 2023 according to the JBI methodology for scoping literature reviews [20,21] in which a pre-established set of steps was followed: (a) the formulation of the research question; (b) the identification of relevant sources of evidence; (c) the selection of sources of evidence for inclusion; (d) data collection/extraction; and (e) grouping, summarisation, and reporting results. The results were reported according to the Preferred Reporting Items for Systematic Reviews and Meta-Analyses Extension for Scoping Reviews (PRISMA-ScR) (Appendix A) [22,23]. The literature review protocol was registered in the Open Science Framework platform [24]. 

### 2.3. Inclusion and Exclusion Criteria

Following the JBI framework for scoping reviews of the literature [25], the team of reviewers collaboratively defined a set of inclusion and exclusion criteria against which they subjected the records obtained from the searches conducted on the different databases and platforms, as explained below:Population—this review considered studies in which the participants were nurses: general care nurses, specialist nurses, midwives, nurse practitioners, and advanced practice nurses. Studies in which the participants were undergraduate nursing students, higher education lecturers, nursing assistants, and auxiliaries were excluded. Studies with participants from different professional groups were included in the dimension of the results that concerned nurses whenever this was unequivocal.Concept—this literature review considered studies focusing on factors with implications for adopting EBP related to professional practice environments and the organisational context of healthcare institutions.Context—this review included studies carried out in different types of professional practice settings, regardless of the sector (public, private or social), the practice setting (primary healthcare, hospital care, public health, occupational health, residential care facilities for the elderly, integrated long-term care units, psychiatry, obstetrics, and palliative care), and the scope of practice (clinical practice, counselling, or management).Types of sources—primary studies and literature reviews and quantitative, qualitative, and mixed methods were considered. Reports or other technical documents relating to the concept of interest were also considered, provided they were issued by professional regulatory bodies, professional associations, scientific societies, or other bodies with recognised authority and standing in the field of EBP. Texts and opinion pieces were excluded, as were protocols, editorials, letters to the editor, short communications, bulletins, and conference abstracts.

### 2.4. Search Strategy

The search strategy, carried out in three stages, as established by the JBI framework for scoping literature reviews [25], sought to locate primary studies, literature reviews, and technical documents published between January 2018 and February 2023 to obtain current and relevant scientific evidence and technical documents directly related to the topic under study. No language restrictions were applied, so it was defined a priori that if there were documents in languages other than English and Portuguese (the authors’ mother tongue), they would first be analysed by their title and abstract in English and only professionally translated if there was a decision to move them on to the full-text reading stage, which was not the case.

An initial search was conducted to locate relevant studies, followed by an analysis of the text and words in the titles and abstracts of significant articles and the indexed terms and keywords used to describe the documents. The words obtained were then used to define an initial search strategy in the MEDLINE database (via EBSCO) to check for evidence that would make it possible to conduct the literature review. In the second stage (1 March 2023), an extended search was carried out in the scientific databases PubMed, Web of Science (Clarivate), CINAHL (via EBSCO), and MEDLINE (via EBSCO) (Appendix A). The search for unpublished studies was conducted on the EThOS, OATD, and RCAAP platforms. An online search was also conducted to locate reports and other technical documents published by professional nursing associations, scientific societies, and others in this field. Finally, in the third stage, the reference lists of all the documents retained for inclusion in the literature review were manually checked against the defined inclusion and exclusion criteria. Table 1 shows the search strategy used in MEDLINE (via EBSCO), which was adapted to the other databases used, adjusting it to their specificities.

### 2.5. Study Selection

The records obtained were exported and uploaded to EndNote^®^ v.20.4 software (Clarivate Analytics, PA, USA) for organisation, analysis, and an initial elimination of duplicates. The process of analysing, sorting, and selecting took place in two separate stages, both conducted on the Rayyan^®^ platform (Qatar Computing Research Institute, Doha, Qatar). Firstly, the records were imported into the Rayyan^®^ platform, where they were subjected to a second check for duplicates, followed by screening by title and abstract, according to the established inclusion and exclusion criteria, by two independent, blinded reviewers. In the second stage, the studies eligible for review were passed on for full-text reading, re-imported into Ryyan^®^, and checked again against the established eligibility criteria, which was also carried out using two independent, blinded reviewers. Reasons for exclusion were standardised and reported. Conflicts between reviewers were resolved through discussion or, if consensus was not possible, with the intervention of a third reviewer. Given the nature of the review and its purpose, the review team decided not to assess the methodological quality of the included studies [26]. The corresponding author was contacted whenever the information available in the paper was insufficient or dubious; if the corresponding author did not reply and the information sought was crucial to the reliability of the information to be extracted, the study was excluded. The flowchart in Figure 1 shows the total number of records identified and the reports included and excluded, indicating the reason for exclusion and the documents included after manually checking the reference lists.

### 2.6. Data Extraction

The data were extracted using a tool designed for this purpose by the authors which was developed in Microsoft Excel^®^ and tested on a random sample of 10 documents to check its clarity and ability to extract relevant data for this study. There was no need to adjust it following the subsequent discussion meeting. For each study included, the following data were extracted: the year of publication, authors, journal name, title, country, clinical practice context in which the study took place (primary studies), type of study, participants or number of documents included (depending on whether it was a primary study or a review study), objective, factors facilitating/promoting the adoption of EBP, factors hindering/inhibiting the adoption of EBP, as well as other aspects that were considered relevant to the review study during the process. Data extraction was carried out independently and blindly by two reviewers.

### 2.7. Data Synthesis and Reporting

A third reviewer aggregated and conformed each pair of independent data extractions into a single document. The data from the included studies were presented using a descriptive narrative supported by tables. The extracted textual elements were subjected to a content analysis, resulting in a coding structure which, through an inductive approach, led to the information being classified, categorised, and linked by similarity and thematic affinity.

The studies were characterised according to the year of publication, the country, and the type of study. When determining the country of origin for each study, the first author’s affiliation was considered, and the countries were then organised and categorised according to their income level [27]. In terms of typology, the studies were organised into two large groups: primary studies (qualitative descriptive studies, Grounded Theory, interpretive description, and intervention methodology) and secondary studies (scoping reviews, integrative reviews, systematic reviews—qualitative, quantitative, and mixed methods, with or without meta-analysis or meta-synthesis—and document analysis). Due to their nature and diversity, the data extracted also made it possible to compare professional practice contexts in terms of factors promoting/hindering the adoption of EBP (e.g., hospitals, primary health care, and long-term care). The reviewers agreed to identify, where appropriate, the names of the instruments used in the primary studies to identify the factors under study. However, this was not considered when registering the review protocol.

## 3. Results

### 3.1. Characterisation of the Reported Studies

After screening and selecting the 2889 records extracted from scientific databases and 11 documents obtained from other sources, it was decided to include 41 documents in the literature review (Figure 1).

As for the chronology of the publications included, nine studies were published in 2018 and 2022, five were published in 2019 and 2021, and thirteen were published in 2020. The studies included were carried out in 21 different countries: twenty-seven in high-income countries (Saudi Arabia, Australia, Belgium, Canada, Cyprus, Denmark, Spain, the United States of America, France, the Netherlands, Ireland, Israel, Norway, Oman, Portugal, and the United Kingdom); five in upper-middle-income countries (China); three in lower-middle-income countries (Iran); and six in low-income countries (Ethiopia, Ghana, and Uganda). Concerning the methodological design of the studies included in the review, 18 were primary studies of a quantitative nature (surveys and cross-sectional, descriptive or correlational studies), 16 were literature review studies (scoping, integrative, systematic—qualitative, quantitative and mixed methods, with or without meta-analysis and meta-synthesis—and documentary analysis), and 7 were primary qualitative studies (qualitative descriptive, Grounded Theory, interpretative description, and intervention methodology) (Appendix A). The Figure 2 shows the temporal and geographical distribution of the documents included in the literature review.

An analysis of the data extracted from the documents included in the review resulted in four main thematic categories, each with at least one set of subcategories, established to organise better and objectify the factors associated with professional practice environments and organisational contexts that have an impact on the adoption of EBP by nurses. Table 2 shows the association between the thematic categories and subcategories and the studies in which they were reported.

### 3.2. Contextual Factors Relating to Organisational Dynamics

#### 3.2.1. Health Organisation Orientation towards EBP

The adoption of a specific EBP implementation model which cuts across the entire organisation is a facilitating factor in the adoption process, acting as an objective guide that is recognised and understood by everyone and which clarifies the healthcare organisation’s position in this area [28,29,30,32,33,38,59]. In addition, healthcare institutions should ensure that EBP implementation processes are carried out by nurses directly involved in providing care [34] as they are better acquainted with any obstacles and are thus better prepared to overcome them, even if they are accompanied by EBP mentors who are not directly linked to the clinical practice context in which the implementation is taking place [30,33], thus promoting the transfer of knowledge and sharing of experiences in terms of EBP adoption [35,59].

Furthermore, health institutions with quality accreditation projects show higher rates of support for the adoption of EBP because they base their orientation and activity on the search for better responses and results in terms of care and management [34,36] and are characterised by environments in which research and innovation are encouraged, with robust clinical governance structures and actual quality policies, thus facilitating the development and implementation of EBP [37].

#### 3.2.2. Organisational Support

Aligning EBP with a health institution’s strategy is fundamental for the organisation’s formal recognition [36,39,40], defining the principles that facilitate change and focusing on the importance of leaders encouraging multi-professional, evidence-based approaches [41]. This dimension of organisational support legitimises the formal authority of clinical services [42,43,44,45,46,50] in leading and affirming the processes of changing practices based on EBP [30,35,47].

Similarly, health institutions that set up expert groups or multi-professional technical working groups [29], with members with EBP skills, to monitor EBP projects developed in clinical services promote the success of EBP compared to health institutions that do not have this organisational unit [28,43,48,49]. In addition, the job description sheets of nurses’ functional content should reflect, in terms of expected competence and desired action, the use of research results in planning activities and the provision of nursing care, giving them the institutional and formal authority to promote the implementation of EBP in clinical practice contexts [40]. In the opposite direction, restructuring processes in organisations and health services which, when they occur, consume considerable energy, resources, and the attention of professionals, in addition to changing the composition of teams, have a profoundly negative impact on ongoing EBP projects and the development of future projects [35].

#### 3.2.3. Organisational Culture

Organisations with a dominant culture geared towards the quality and safety of nursing care, with adequately established internal control and result evaluation systems, tend to favour the implementation of EBP [51], showing less resistance to change and encouraging nurses to challenge professional practices and established behaviours by actively looking for better alternatives [28,35,38,52]. In this respect, an organisational culture that values research and research results to improve care will facilitate the implementation and dissemination of the processes inherent in EBP [37,53].

Organisational culture and its orientation towards EBP cannot be dissociated from the size of healthcare organisations, considering that in large hospitals that serve very diverse audiences, barriers of a cultural nature can emerge, such as those arising from the socialisation process within teams. In contrast, in smaller hospitals with a propensity to create dynamics of greater proximity and affinity, conditions can be made that facilitate the adoption of EBP precisely because of the fluidity and ease of transferring and sharing knowledge and experiences [54].

#### 3.2.4. Training and Professional Development

Nurses believe that healthcare organisations should promote access to specific training in EBP—postgraduate or short-term [55,56]—including incentives for training [40] or internal training promoted by the healthcare institutions themselves [34]. Continuous training aimed at EBP should be taken on as a central dimension in the organisation with a view to professional development, which is seen as a determining factor in the emergence of environments that facilitate EBP [51]. In cases in which local EBP training initiatives are chosen, they should be promoted in stages, gradually introducing nurses to the concepts, methods, and processes inherent in adopting EBP [33]. The creation of favourable conditions for disseminating the results of EBP implementation through participation in congresses, conferences, communications, and scientific meetings, with the publication of articles and the presentation of posters and oral communications, is also associated with consolidating this approach [51].

#### 3.2.5. Articulation with External Organisations

The support of external organisations in providing the resources (material, human, and technical) needed to implement EBP when health institutions cannot do so on their own emerges as a facilitating aspect of the process [30,55]. It is worth highlighting the establishment of partnerships with higher-education institutions to promote the sharing of experience and knowledge and thus prepare nursing teams for their progressive autonomy in implementing scientific evidence [29,39,40,43,51]. In this area, it is also essential to consider the need for health policymakers to prioritise EBP in the sector and to improve the quality, safety, and cost-effectiveness of care by legislating, regulating, and issuing strategic and technical guidelines to encourage the adoption of EBP [55].

### 3.3. Contextual Factors Relating to Management and Leadership

#### Nurse Managers and Nursing Leadership

The role of nurse managers is crucial in promoting EBP in healthcare institutions [42,51,52,54,57,58,59], defining, at each management level or department, the policy and orientation of the respective services in terms of the quality and safety of nursing care [41,60], mainly through a collective team vision of the importance of adopting EBP [36,38,39].

Nursing managers must take on the adoption of EBP as a central dimension in defining their priorities and in the strategic orientation of the nursing service at the institutional level [28,33], considering that the less leadership and orientation there is, the more obstacles nurses will encounter in adopting EBP [54] since it is known that the formal authority of the nurse manager legitimises the implementation of EBP [41,61]. Nurse managers also act as role models and mentors [36]. They can set an example when it comes to using evidence [62,63], empowering teams [36], sharing experiences and knowledge, supervising processes [34,63], promoting spaces for wide-ranging discussion [29], and recognising merit whenever it arises [34,51]. They mediate interprofessional conflicts [62] and facilitate communication and teamwork [41,51,63,64].

### 3.4. Contextual Factors Relating to Teamwork and Communication

#### Communication and Peer Relations

Resistance to change on the part of nurses in teams and other professional groups, including doctors, is a solid inhibitor to the implementation of EBP [9,42,43,45,46,47,48,50,61,65,66]. Professional practices based on old habits and traditions that are not questioned or challenged severely hamper initiatives that seek to promote the integration of the best scientific evidence into care practice [29].

Nurses who promote EBP initiatives commonly do not effectively communicate the relevance of their actions within their teams, showing insufficient and inadequate communication [42,53]. This difficulty in communicating and making EBP relevant and the need for change prevail is also felt in the relationship established with nurse managers, resulting in damage [34,41,60].

### 3.5. Context Factors Relating to Resources and Infrastructure

#### 3.5.1. Human Resources

Staffing is a vital resource for implementing EBP, i.e., with enough nurses, it is possible to conciliate the provision of nursing care with the processes inherent in implementing EBP [9,40,56]. This impossibility stems from the work overload that nurses are already subjected to [32,56,66] and high internal turnover [61], which generates professional dissatisfaction and accentuates the unwillingness to implement EBP, which translates into a lack of commitment and motivation [51,55,62].

#### 3.5.2. Time

Nurses point out that they do not have enough time to carry out the processes of researching, locating, reading and analysing scientific literature, primarily as a result of understaffing, prioritising the provision of care to the detriment of EBP [9,29,30,35,39,40,42,43,44,45,46,48,50,57,59,61,65,67,68,69], resulting in the fact that if they want to develop EBP projects, they have to carry them out outside of working hours, i.e., at home, compromising their personal rest time [57].

#### 3.5.3. Adequacy and Availability of Infrastructure

The number of computers available to accommodate the administrative demands associated with the provision of care as well as the activities inherent to EBP, but also the type and speed of internet access, are identified as determining factors for the success of EBP [39,40,47,52,57,66]. The existence of suitable physical conditions for the development of EBP, such as meeting rooms, desks and chairs in spaces exclusively dedicated to this activity [42,55], and even libraries [51,65], are also aspects that directly impact the implementation of EBP. Finally, the physical conditions of clinical practice settings, which make it impossible to implement the changes that emerge from applying scientific evidence [40,45,50,51], are also signalled as factors that inhibit EBP due to their insufficiency or inadequacy.

#### 3.5.4. Material and Other Resources

Factors associated with the adoption of EBP in this subdomain include the need to provide nurses with up-to-date documentary collections that are appropriate to the nature and specificity of clinical practice contexts, as well as access to relevant scientific databases to locate current, high-quality research results [9,40,51,55,66]. On the other hand, access to modern materials and equipment, including clinical consumables and diagnostic and therapeutic aids, is a highly relevant factor in implementing guidelines and recent evidence in specific care contexts [51,54,56].

### 3.6. Specificities Inherent to the Context of Clinical Practice

There were no noteworthy differences in the factors associated with implementing EBP in the different types of clinical services covered by the studies included in this review. Notwithstanding the above, some aspects are worth mentioning due to the specificity of the contexts in which they were identified. For example, there is a more significant reference to the limitations of time and the availability of human resources in integrated long-term care, palliative care, and primary health care, with an impact on the adoption of EBP by nurses, resulting in the impossibility of ensuring a capable response in terms of nursing care at the same time as the development of EBP implementation processes [48,61]. Furthermore, the day-to-day management of existing material resources in long-term care (which are often insufficient) is a challenge, which is why, in the opinion of nurses, the use of EBP only intensifies this difficulty because it forces them to fulfil quality standards that are impossible to achieve in a context of scarcity [61]. Also, in this area, and in conclusion, nurses draw attention to staff turnover and the impact this has on team stability, a fundamental condition for the solid development of EBP implementation projects which does not affect all services with equal intensity and coverage, mainly services with a higher level of criticality (e.g., operating theatres, emergency services, or intensive care units), which seem to be more protected from turnover due to the high specialisation of nurses [35,61].

### 3.7. Specificities Inherent to Geographical or Geopolitical Contexts

In line with the specificities arising from the context of clinical practice, no significant differences were identified in the factors associated with different countries, depending on their income group. Nevertheless, it is important to emphasise certain factors that significantly impact the adoption of EBP by nurses in lower-income countries.

Although the broad recognition of limitations of a material and infrastructural nature, including the availability of computer equipment, internet access and speed, access to and availability of sources of information, particularly electronic databases, but also the existence of infrastructure and material resources to implement EBP processes and evidence itself, were transversal to all the studies included, these limitations are substantially more emphasised in low- and lower-middle-income countries [30,39,40,42,47,52,53,66], with all that this represents in terms of EBP adoption. The same is true of nurses’ formal authority to adopt EBP and change professional practices, both among peers and especially in the face of a lack of recognition from doctors [40,42,47,52,53,66]. Finally, and acknowledging various shortcomings in terms of knowledge and experience in implementing EBP which stem from its specificity and the context in which it is located, studies from these countries highlight the need for non-governmental organisations, as well as organisations and higher-education institutions located in countries with a greater availability of resources, to provide support of various kinds, including the sharing of know-how, to develop, strengthen, and consolidate EBP as a central approach to the design and provision of nursing care [30].

### 3.8. Instruments Used to Identify the Contextual Factors Influencing the Adoption of EBP

Although it was not so much an objective as a research question, during the data analysis, extraction, and synthesis work, it seemed pertinent to the team of reviewers to identify the data collection instruments, whenever possible, used to determine the factors associated with the professional practice context and the organisational context involved in implementing EBP. Thus, of the 25 primary studies included in this literature synthesis, 23 referred to the instruments used in them. Of these, due to the nature of their methodological design, six were semi-structured interviews (some of which included non-participant observation), three were instruments authored by the authors (with no name given to the instrument), and another three were partial adaptations of instruments already validated for different contexts and populations but not subject to new validation within the scope of the studies for which the adaptation was intended.

The most used instrument was the “Barriers Research Utilisation Scale”, present in six included primary studies [42,44,45,46,50,58], followed by the “Implementation Climate Scale” [63], the “Alberta Context Tool [64]”, and the “Facilitators to Research Utilisation Scale” [58] with only one use, the “Nursing Evidence-Based Practice Survey” [29], the “Organisational Culture and Readiness for System-Wide Integration of Evidence-Based Practice Scale” [38], the “Evidence-Based Practice Nursing Leadership Scale” [34], the “Evidence-Based Practice Work Environment Scale” [34], and the “Developing Evidence-Based Practice Questionnaire” [60].

## 4. Discussion

The results obtained made it possible to identify four broad categories of factors and a set of subcategories that answer, on the one hand, the research question that led to the literature review and, on the other, the fundamental objective defined for this study. The categories and subcategories established, applied to the design of an intervention project aimed at promoting environments that facilitate EBP, themselves configure intervention domains and subdomains for which strategies with a high degree of specificity can be designed and implemented, also contributing to a better understanding of the relevance of organisations and professional practice contexts in the implementation of EBP [70]. The findings underscore the pivotal role of both the organizational context and the context of professional practice in shaping the landscape of EBP implementation by nurses. This extends beyond considering EBP merely within the scope of individual nurses’ knowledge and competence, highlighting its integration into the broader organizational sphere. The results emphasize the imperative for interventions at the organizational level to ensure the effective recognition of EBP as a pertinent and indispensable component of professional practice for nurses and all healthcare professionals across diverse health organizations. The emphasis on organizational dynamics in the implementation of EBP reaffirms the interconnectedness of individual competencies and organizational support, emphasizing the need for a comprehensive approach to foster a culture in which EBP is acknowledged and seamlessly integrated into the fabric of healthcare practices.

Generally speaking, the factors associated with professional practice contexts and organisational dynamics, as far as the conditions for adopting EBP are concerned, have remained relatively unchanged over the last few years [50], a circumstance that is surprising; given that the situations and problems have been well identified, and that EBP remains an apparent priority for healthcare organisations, this scenario should already have been, if not wholly, at least partially modified [71,72].

The factors that stem intrinsically from organisations have emerged as the most significant in terms of their impact on the adoption of EBP by nurses [73,74]. From its strategic orientation, it is up to an organisation to define its commitment to implementing EBP as a structuring approach for its quality, safety, and cost-effectiveness policy [12,15]. Suppose a particular measure is a priority for an organisation. In that case, if it is established by the board of directors or the nursing department, and if this option is also clearly communicated, the employees must align themselves with it, integrating the directive and acting accordingly [75,76].

In the specific case of EBP, the existence of an institutionally established implementation model that is easy for clinical professionals to assimilate, in which the associated processes and procedures have been presented, discussed, and reviewed through training initiatives, is a way of institutionally assuming EBP as a strategic priority and laying the foundations for its adoption [15,77,78]. This level of definition and clarity is generally better established in organisations involved in accreditation and certification programmes (e.g., Magnet Recognition Program, CHKS, or Joint Commission), insofar as the implementation of EBP is already monitored by external bodies that verify and validate internal quality processes or, in some way, the quality criteria they establish oblige healthcare organisations to resort to procedures that essentially involve using EBP to update internal procedures, algorithms, standards of clinical practice, and manuals of good practice [79,80,81].

Aside from an organisation’s willingness and readiness to promote EBP, it is also essential that the necessary resources are made available, starting with the provision of spaces and equipment exclusively for this purpose: spaces in which the implementation teams can meet, discuss, and assess the relevance and priority of different projects [45,50,55]. In healthcare organisations where EBP is still in its infancy and is not particularly well structured, it may make sense to establish teams and groups of experts with expertise in EBP, including members of the organisation itself (where these exist) but also from external partners, such as universities, professional associations, or scientific societies [82]. This option could lead to the progressive expansion and consolidation of EBP in clinical practice contexts as it promotes the development of competence through the supervision and mentoring of the professionals involved in the processes at the local level [83].

Another crucial aspect, but on which is challenging to address and resolve, is the need for more receptiveness of teams to change [84]. Healthcare organisations may be willing to promote EBP in clinical practice contexts. However, professionals, particularly nurses, may not be willing to do so and may not feel the need to question and challenge their practices. Healthcare organisations need to take a different approach, focusing on raising awareness among teams [55,85]. This awareness raising must necessarily involve nurse managers, who are responsible for local coordination of services and teams but also for implementing, at an operational level, using the strategies they consider most appropriate and adapted to each context, the determinations that result from the strategic orientation of the boards of directors of health institutions [86]. An essential part of this awareness raising must focus on the quality and safety of the nursing care provided, not only from the perspective of the person, group, or community being cared for but also from that of the health professional, creating the conditions for safer and more responsible care practice [87,88].

As far as the role of nurse managers is concerned, particularly in promoting EBP, it continues after carrying out senior management’s directives [84,89]. Their role in motivating teams, encouraging and recognising merit, facilitating communication, and resolving conflicts, particularly those with other professional groups, is essential [90,91,92]. Capable leadership from nurse managers gives the nurses on these teams the formal authority to lead their projects to implement evidence and change professional practices [93,94]. Nurse managers also play a fundamental role in monitoring and mentoring the implementation of evidence, providing support for the process [84,95,96].

In organisations and professional practice contexts in which EBP is still in its infancy, it is necessary to be aware of the possible need to make a very considerable initial investment at various levels even before any results are achieved [39,97]. This investment involves investing in the teams’ size and suitability, training, and preparation for EBP and adapting the physical infrastructure and information systems to implement EBP [49,98]. Modern healthcare organisations are also centres of knowledge, so the modernisation of libraries and access to scientific databases, even through protocols and cooperation agreements with partner higher education institutions, needs to be considered if the intention is to consider implementing EBP [8,99] seriously.

In conclusion, and with relative certainty, it can be assumed that it does not make much sense to demand that nurses base their care practice on the best scientific evidence available when the minimum conditions necessary for this are not met by health institutions and clinical services, especially those related to the existence of a sufficient number of nurses to reconcile, without conflict and harm to either party, the provision of nursing care with the activities that result from the implementation and development of EBP. Nor does it make sense to transfer the responsibility for implementing EBP to the dimension of nurses’ professional responsibility, requiring them to acquire and consolidate EBP competencies, since for these to be expressed in context, to be consolidated and to result in real and objective gains, both for health systems and services and for the recipients of health care, healthcare organisations must take on EH&S and create the necessary conditions (those that depend on them) for its implementation, recognising their decisive role in facilitating the implementation of EH&S from the outset, orienting their strategy towards the continuous improvement of quality of care and making this orientation unequivocal for the entire organisation.

### 4.1. Implications for Nursing Management and Research

From this scoping review of the literature, the factors related to the development and implementation of EBP arising from professional practice environments and the organisational context in healthcare institutions remain relatively unchanged, i.e., in this literature review. However, it has proposed and presented an organisation of factors in a structure of categories and subcategories different from others already presented, converging with findings already known by the scientific and professional community from the various primary studies included.

A fundamental question arises from this which is related to why little has changed over the years in the contexts of clinical and organisational practice, particularly in which modifiable factors, such as management support for EBP and, in a particular way, nurse managers (with increased responsibility in this area as those responsible at an operational level for health services), in every way compromise, for example, the formal authority of nurses who implement projects to integrate scientific evidence into care practice and change professional practices. Perhaps it would make sense to shift the focus of the research away from the aspects primarily associated with the individual dimension of nurses and health professionals in terms of preparation for the adoption of EBP and focus the study on the reasons behind the barriers related to professional and organisational practice environments remaining relatively unchanged, exploring the dissonance that exists between the dominant discourse that values and defends the adoption of EBP and the action that is slow to produce practical effects in terms of its implementation.

Therefore, it is also important to reflect deeply on why organisations and health services, including nursing managers and teams, remain obstacles to EBP. It is also essential, in terms of disseminating the results of EBP, to highlight the gains and results obtained, from those that affect those targeted in nursing care to the efficiency gains seen by health services and systems, using robust and unequivocal indicators, even though it must be assumed that in the initial phase of implementing EBP, the results may be meagre. It is necessary to correctly scale the resources allocated to implementing EBP.

To instigate a transformative shift in applying EBP, it is imperative to advocate for consistent and comprehensive strategies. While various pedagogical approaches have demonstrated efficacy in preparing nurses for EBP implementation—ranging from small group exercises, critical analyses of articles, and case studies to bibliographic research and simulated practice—it is crucial to recognize that embracing EBP as a scientific methodology is markedly different from effectively putting it into practice. The challenge lies not solely in the individual healthcare professional’s willingness or competence but extends to a multifaceted array of factors woven into professional and organizational practice contexts. Given the intricate nature of EBP, a compelling argument can be made for a strategy that transcends individual training efforts. Instead, it is vital to illuminate the significance of EBP to key stakeholders in health organizations and decision-making processes. This entails conveying the theoretical underpinnings and substantiating the practical benefits through visible demonstrations. A persuasive tool is a tangible exhibition of the positive impacts on health outcomes, quality, and safety in nursing care. This demonstration should extend to the quantifiable satisfaction of patients whose treatments are rooted in EBP, with a direct correlation between evidence-based approaches and patient contentment. Moreover, this approach aims to also showcase the professional satisfaction of nurses, illustrating how adherence to EBP enhances their practice. Beyond the individual level, the strategy seeks to underline the broader advantages, including gains in efficiency and cost savings. By optimizing costs and discarding outdated and ill-suited professional practices that contribute little to patient outcomes, such an approach not only minimizes the waste of resources but also mitigates risks for patients and health organizations. In essence, a tangible exhibition of the positive impacts of EBP endeavours to create a compelling narrative around EBP that transcends theoretical acceptance, fostering a culture in which evidence-based approaches become ingrained, celebrated, and integral to the overarching success of healthcare delivery.

Finally, it should be noted that one of the lines of research to be considered on this topic could focus on the design and conception of studies comparing the results of nursing practices and interventions informed by EBP to conventional practices, for example, those handed down by tradition, highlighting the gains obtained in terms of resolving clinical conditions, the quality and safety of care, the cost-effectiveness of nursing care, and patient and nurse satisfaction. From here, a set of indicators capable of objectifying these gains and making them visible should also be defined.

### 4.2. Limitations

The main limitation of this literature review, apart from those inherent to the method chosen (e.g., the lack of a formal quality assessment of the included studies, the limited synthesis of evidence with no quantitative or statistical analysis, and the heterogeneity of the included study designs), is the proposed structure of the categories and subcategories used to organise the findings of the review insofar as it results from the vision and interpretation of the reviewers so that other researchers, with a different point of view and different experience in the field of EBP, could have proceeded differently in this area.

It was also a limitation to report that the methodological quality of the studies was not assessed, even though this procedure is optional in this type of literature review design. This choice was mainly due to the intention of exploring and mapping factors within the framework of the defined concept of interest using the most comprehensive number of sources possible, so if an assessment of the methodological quality of the studies had been carried out, several would have been excluded, thus limiting the diversity of sources that shape the results of this study.

Despite the relative diversity of countries in which the studies included were conducted, especially the primary studies, the results obtained, without prejudice to mapping and organising factors inherent to the professional practice environment and organisational nature in terms of their impact on the implementation of EBP, should be analysed with the caution that results from the geographical distribution of the studies included in the review, and any temptation to generalise should be avoided. Although this is a limitation, it is also a virtue. It has shown that, despite the considerable and well-known differences between countries in their income levels, the same reality is shared in this area, albeit to varying degrees.

## 5. Conclusions

The comprehensive analysis of the studies incorporated in this literature review has illuminated critical facets influencing nurses’ adoption of EBP. The persistent and relatively unchanged barriers posed by professional practice environments and organisational contexts underscore the imperative for transformative initiatives. The key takeaway is that successfully integrating EBP necessitates a broader commitment from healthcare organisations, involving stakeholders at all levels, from board directors to operational managers.

EBP should not be merely a theoretical framework but an integral component of the strategic orientation of healthcare organisations. Recognising EBP as a critical success factor in enhancing the quality and safety of care is paramount. This necessitates a paradigm shift in which the implementation of EBP is ingrained in the organisational culture.

Furthermore, beyond logistical and technical requisites, responsibility also lies with those advocating for EBP to construct robust indicators (encompassing structure, process, and outcome measures). These indicators serve as tangible evidence of the value and benefits derived from the application of EBP. Effective communication of these indicators to decision makers and healthcare team members becomes imperative, fostering a collective understanding of the merits of EBP.

In the broader context, this review and similar studies play a pivotal role in shedding light on the persistent gap between intention and implementation in EBP. By providing objective insights into the challenges and potential solutions, these findings serve as a foundation for developing targeted strategies. The results of this review study will be instrumental in formulating a comprehensive strategy to create conducive conditions for the widespread adoption of EBP in clinical services and healthcare organisations. Ultimately, this strategic approach aims to bridge the existing divide between intention and action, fostering a culture in which EBP becomes an integral and effective component of professional nursing practice.

## Figures and Tables

**Figure 1 healthcare-12-00245-f001:**
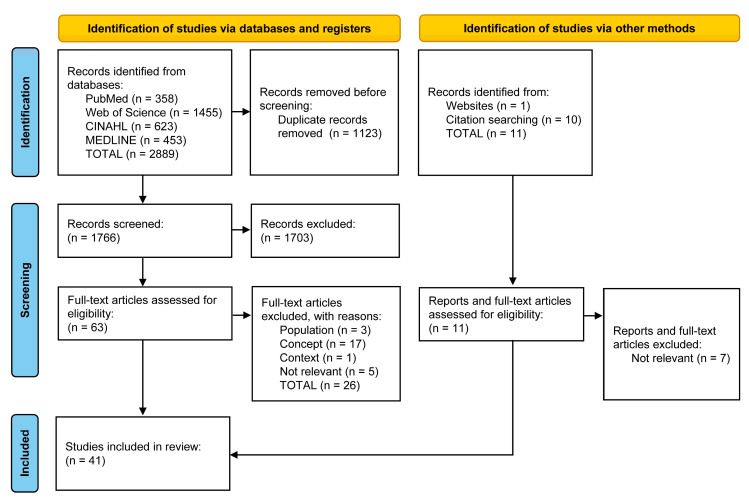
PRISMA-ScR flowchart for identifying, screening, and selecting the articles included in the scoping review.

**Figure 2 healthcare-12-00245-f002:**
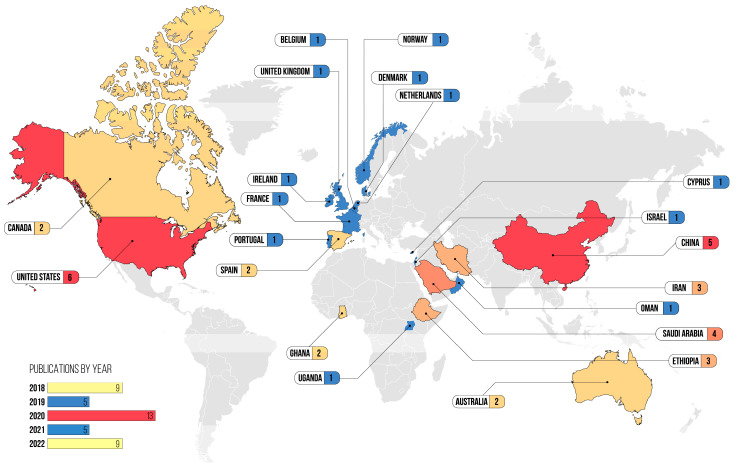
Chronological and geographical distribution of the studies included in the literature review (figure by the first author).

**Table 1 healthcare-12-00245-t001:** Search strategy used in MEDLINE (via EBSCO).

Search No.	Search Terms and Expressions	Results
S1	MH “Nurses” OR TI “Nurs*” OR AB “Nurs*”	526,266
S2	MH “Students, College” OR MH “Students, Nursing” OR MH “Students, Pre-Nursing” OR TI “Undergraduat*” OR AB “Undergraduat*” OR TI “Student*” OR AB “Student*” OR TI “nursing student*” OR AB “nursing student*”	377,263
S3	S1 NOT S2	479,521
S4	AB “facilitator” OR AB “enabler” OR AB “enhancer” OR AB “implement*” OR AB “helper”	744,469
S5	AB “hindering” OR AB “obstacle” OR AB “barrier” OR AB “difficult*” OR AB “impediment*”	981,126
S6	S4 OR S5	1,677,935
S7	MH “Professional Practice, Evidence-Based” OR TI “evidence-based practice” OR AB “evidence-based practice” OR TI “EBP” OR AB “EBP”	19,246
S8	S6 AND S7	6718
S9	AB “health organization*” OR AB “healthcare organization*” OR AB “healthcare organization*” OR AB “hospital*” OR AB “organization*” OR AB “organization*”	1,771,592
S10	S3 AND S8 AND S9	453

**Table 2 healthcare-12-00245-t002:** Categories and subcategories of factors associated with the adoption of EBP and the studies in which they were identified.

Category	Subcategory	Study
Contextual factors relating to organisational dynamics	Health organisation orientation towards EBP	[28,29,30,31,32,33,34,35,36,37,38]
	Organisational support	[28,29,30,35,36,39,40,41,42,43,44,45,46,47,48,49,50]
	Organisational culture	[28,35,37,38,51,52,53,54]
	Training and professional development	[33,34,40,51,55,56]
	Articulation with external organisations	[29,30,39,40,43,51,55]
Contextual factors relating to management and leadership	Nurse managers and nursing leadership	[28,29,33,34,36,38,39,41,42,51,52,54,57,58,59,60,61,62,63,64]
Contextual factors relating to teamwork and communication	Communication and peer relations	[29,34,41,42,43,45,46,47,48,50,53,60,61,65,66]
Context factors relating to resources and infrastructure	Human resources	[9,32,40,51,55,56,61,62,66]
	Time	[29,30,35,39,40,42,43,44,45,46,48,50,57,59,61,65,67,68,69]
	Adequacy and availability of infrastructure	[39,40,42,45,47,50,51,52,55,57,65,66]
	Material and other resources	[9,40,51,55,56,66]

## Data Availability

Not applicable.

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
