# Peer review of "Exploring Professional Practice Environments and Organisational Context Factors Affecting Nurses’ Adoption of Evidence-Based Practice: A Scoping Review"

_healthcare, 2024, doi:10.3390/healthcare12020245_

Round 1

Reviewer 1 Report

Comments and Suggestions for Authors

Mauscript on a high level. It presents an important topic for the field of nursing which is EBP. The authors analyzed a long period of time (2018-2023) which is research reliability. The work is written in an understandable language. It comprehensively presents the topic.

The main reservation I have is about the conclusions. They represent a repetition of the results. Please rewrite the conclusions and write out specific (point) conclusions.

It would be worth considering a subsection on ideas on how to spread EBP and EBM among the nursing peresonnel.

Author Response

Dear Reviewer,

Thank you sincerely for your meticulous analysis of our manuscript and for providing invaluable insights and recommendations. We greatly appreciate the time and effort you have dedicated to enhancing the quality of our work.

We have carefully considered your feedback and made significant revisions to the conclusion chapter, aligning it with your suggestion to emphasize the necessity for a paradigm shift in the effective implementation of Evidence-Based Practice (EBP). We have reframed the focus and language, ensuring that it complements rather than duplicates the findings presented in the results chapter.

Regarding your suggestion to include a subsection on disseminating EBP and EBM among nursing staff, we acknowledge the complexity of this topic and concur that it could indeed warrant a dedicated literature review. To address this, we have opted to delve deeper into the analysis provided in subsection "4.1. Implications for nursing, management, and research." We believe that this approach will offer a more nuanced exploration of the implications and potential strategies for advancing EBP among nursing professionals.

We are eager to hear your thoughts on these revisions and whether they align with your expectations. Your guidance has been instrumental in refining our manuscript, and we want to ensure that our modifications accurately address your suggestions.

Once again, we express our gratitude for your thorough review and constructive feedback. Your commitment to the scholarly review process has undoubtedly enriched the content of our article.

All the best,

The authors.

>>>>>>>>>> Conclusion <<<<<<<<<<

The comprehensive analysis of the studies incorporated in this literature review has illuminated critical facets influencing nurses' adoption of EBP. The persistent and rela-tively unchanged barriers posed by professional practice environments and organisa-tional contexts underscore the imperative for transformative initiatives. The key takeaway is that successfully integrating EBP necessitates a broader commitment from healthcare organisations, involving stakeholders at all levels, from board directors to operational managers.

EBP should not be merely a theoretical framework but an integral component of the strategic orientation of healthcare organisations. Recognising EBP as a critical success factor in enhancing the quality and safety of care is paramount. This necessitates a paradigm shift where the implementation of EBP is ingrained in the organisational culture.

Furthermore, beyond the logistical and technical requisites, the responsibility also lies with those advocating for EBP to construct robust indicators (encompassing structure, process, and outcome measures). These indicators serve as tangible evidence of the value and benefits derived from the application of EBP. Effective communication of these in-dicators to decision-makers and healthcare team members becomes imperative, fostering a collective understanding of the merits of EBP.

In the broader context, this review and similar studies play a pivotal role in shedding light on the persistent gap between intention and implementation in EBP. By providing objective insights into the challenges and potential solutions, these findings serve as a foundation for developing targeted strategies. The results of this review study will be instrumental in formulating a comprehensive strategy to create conducive conditions for the widespread adoption of EBP in clinical services and healthcare organisations. Ulti-mately, this strategic approach aims to bridge the existing divide between intention and action, fostering a culture where EBP becomes an integral and effective component of professional nursing practice.

>>>>>>>>>> Spread EBP and EBM among the nursing peresonnel <<<<<<<<<<

To instigate a transformative shift in applying EBP, it is imperative to advocate for consistent and comprehensive strategies. While various pedagogical approaches have demonstrated efficacy in preparing nurses for EBP implementation – ranging from small group exercises, critical analysis of articles, and case studies to bibliographic research and simulated practice – It is crucial to recognize that embracing EBP as a scientific method-ology is markedly different from effectively putting it into practice. The challenge lies not solely in the individual healthcare professional's willingness or competence but extends to a multifaceted array of factors woven into professional and organizational practice con-texts. Given the intricate nature of EBP, a compelling argument can be made for a strategy that transcends individual training efforts. Instead, it is vital to illuminate the significance of EBP to key stakeholders in health organizations and decision-making processes. This entails conveying the theoretical underpinnings and substantiating the practical benefits through visible demonstrations. A persuasive tool is a tangible exhibition of the positive impacts on health outcomes, quality, and safety in nursing care. This demonstration should extend to the quantifiable satisfaction of patients whose treatments are rooted in EBP, with a direct correlation between evidence-based approaches and patient con-tentment. Moreover, this approach aims also to showcase the professional satisfaction of nurses, illustrating how adherence to EBP enhances their practice. Beyond the individual level, the strategy seeks to underline the broader advantages, including gains in effi-ciency and cost savings. By optimizing costs and discarding outdated and ill-suited professional practices that contribute little to patient outcomes, such an approach not only minimizes waste of resources but also mitigates risks for patients and health organiza-tions. In essence, a tangible exhibition of the positive impacts of EBP endeavours to create a compelling narrative around EBP that transcends theoretical acceptance, fostering a culture where evidence-based approaches become ingrained, celebrated, and integral to the overarching success of healthcare delivery.

Reviewer 2 Report

Comments and Suggestions for Authors

Thank you for allowing me to review this paper.Overall, it is an interesting area of study with nurses' adoption of evidanced-based practice. I hope the following comments help the author prepare the revised manuscript.

1.     Abstract: I think that the keywords should be drawn from the list of MeSH (Medical Subject Headings) keywords.

2.     Method section: There are some problems in the research design.

(1)    How do you consider indicators for deciding between a systematic review and a scoping review approach?

(2) Why do you select 2018-2023 for this scoping review?

3.     Results section: The results section is very confusing for me. I suggest having a table that categorizes it into main information and subcategories linked by similarity and thematic affinity. Please carefully recheck the definitions of the categories and subcategories.

Author Response

Dear Reviewer,

Thank you sincerely for your meticulous analysis of our manuscript and for providing invaluable insights and recommendations. We greatly appreciate the time and effort you have dedicated to enhancing the quality of our work.

We think there may have been some confusion regarding the keywords since all the keywords presented in the abstract were extracted from the MeSH list.

Concerning the research design, in the last section of the introductory chapter, we justify why we opted for a scoping review rather than a systematic literature review, supporting this decision on the assumptions defined by Munn et al. 2018. Indeed, this option stems from the heterogeneity of the sources of evidence included in the review (which is not compatible with a systematic literature review) but also from the aim of mapping and organising factors and areas of intervention, identifying in the process any knowledge gaps that may present opportunities for future research.

As for the time period used, we only intended to look at "current literature", which is why the period 2018-2023 was defined. We only wanted to systematise the most current literature on the topic.

We may not fully understand the question you asked in (3). In fact, there is a table (Table 2) that systematises the main categories and subcategories defined within each one. The subcategories are associated with the respective main categories by similarity and thematic affinity. For example, the main category "Context factors relating to resources and infrastructure" contains four subcategories, all of which relate to different types of resources/infrastructure (e.g. "Human resources", "Time", "Adequacy and availability of infrastructure", and "Material and other resources"). The process of analysing and defining the categories and subcategories was exhaustive: initially, only the main categories emerged; however, due to the diversity of "themes" within most of the main categories, and to better organise and generate a better understanding of the factors, we decided to define subcategories, restricting the scope of each one, in essence making them more objective.

In any case, we admit that we may not have understood your doubt/recommendation, so we ask you if the justifications we have given clarify your doubts or if you could better clarify what you wanted to see covered in the manuscript so that we can also understand the feasibility of doing so, since at this point, with all the content analysis work having been completed, a substantial change to the categories may not be feasible.

We are eager to hear your thoughts on these revisions and whether they align with your expectations.

Once again, we express our gratitude for your thorough review and constructive feedback.

Kind regards,

The authors.

Reviewer 3 Report

Comments and Suggestions for Authors

Dear Authors,

congratulations on the topic and for the methodological conduct of your study. Your manuscript can have an impact on healthcare organisations, as it increases knowledge about the factors and obstacles involved in adopting EBP in clinical practice. I have only some minor suggestion:

- You mentioned the registered trademark (®) of the Rayyan platform (line 180), however previously it was not done. Perhaps it would be correct to do it also for the EndNote software (line 172), consistent with what was done for Rayyan. You may specify the registered trademark only the first time you mention a software or intellectual property, without doing so later, as with Excel (line 194).

- This sentence should be summarized as a limit in the corresponding paragraph (4.2): "The correspondence author was contacted whenever the information available in the paper was insufficient or dubious; if the correspondence author did not reply and the information sought was crucial to the reliability of the information to be extracted, the study was excluded." (lines 185-88). I cannot see in the PRISMA Flowchart where are the studies excluded due for failure to reply by the author you contacted.

-Please specify the hidden methodological limits when you say "apart from those inherent to the method chosen" (lines 576-77)

-Please, explain better the meaning of "Although this is a limitation, it is also a virtue. It has shown that, despite the considerable and well-known differences between countries in their income level, the same reality is shared in this area, albeit to varying degrees." (lines 593-95). If possible, please support with bibliographic references.

- This sentence could be a repetition of what was previously expressed in the results: "A total of 41 studies published in countries on six continents between 2018 and 2022 were considered eligible for this literature review." (lines 598-99).  It might be useful to use another formula to introduce conclusions, or else you might delete it.

Author Response

Dear Reviewer,

Thank you sincerely for your meticulous analysis of our manuscript and for providing invaluable insights and recommendations. We greatly appreciate the time and effort you have dedicated to enhancing the quality of our work.

A registered trademark (®) has been inserted in all references.

Concerning the issue of requesting additional clarification from authors whose articles were unclear, we would like to clarify this issue in two distinct dimensions. This paragraph is inserted to reinforce our commitment to methodological rigour and the robustness of our present results. It does not constitute a study limitation, which brings us to the second dimension we wish to clarify. The PRISMA and PRISMA-ScR flowchart doesn't include any branch relating to exclusion due to "lack of response from the authors when asked about the studies"; the studies, even those that were rejected due to lack of response from the respective authors, were excluded because they failed to clarify the fundamental reasons for exclusion ("population", "concept", "context" and "not relevant").

Concerning the phrase "apart from those inherent to the method chosen", we have inserted some of the limitations referred to in the text.

Regarding the doubt related to the sentence, "Although this is a limitation, it is also a virtue. It has shown that, despite the considerable and well-known differences between countries in their income level, the same reality is shared in this area, albeit to varying degrees." The justification for this can be found in the sentence that immediately follows: "It has shown that, despite the considerable and well-known differences between countries in their income level, the same reality is shared in this area, albeit to varying degrees." The references that support this assumption, as well as the findings that substantiate it, are identified in section "3.7. Specificities inherent to the geographical or geopolitical context" of the manuscript.

Regarding your comment on the sentence in the conclusion chapter, "A total of 41 studies published in countries on six continents between 2018 and 2022 were considered eligible for this literature review.", we would like to inform you that the entire conclusion chapter has been reworded.

Once again, we express our sincere appreciation for your valuable contribution to our article. Your insights have been instrumental in refining our work, and we look forward to your continued guidance.

All the best,

The authors.

>>>>>>>>>> Scoping review limitations <<<<<<<<<<

[…] chosen (e.g. lack of formal quality assessment of the included studies, limited synthesis of evidence with no quantitative or statistical analysis, heterogeneity of the included study designs) […].

>>>>>>>>>> Conclusion <<<<<<<<<<

The comprehensive analysis of the studies incorporated in this literature review has illuminated critical facets influencing nurses' adoption of EBP. The persistent and rela-tively unchanged barriers posed by professional practice environments and organisa-tional contexts underscore the imperative for transformative initiatives. The key takeaway is that successfully integrating EBP necessitates a broader commitment from healthcare organisations, involving stakeholders at all levels, from board directors to operational managers.

EBP should not be merely a theoretical framework but an integral component of the strategic orientation of healthcare organisations. Recognising EBP as a critical success factor in enhancing the quality and safety of care is paramount. This necessitates a paradigm shift where the implementation of EBP is ingrained in the organisational culture.

Furthermore, beyond the logistical and technical requisites, the responsibility also lies with those advocating for EBP to construct robust indicators (encompassing structure, process, and outcome measures). These indicators serve as tangible evidence of the value and benefits derived from the application of EBP. Effective communication of these in-dicators to decision-makers and healthcare team members becomes imperative, fostering a collective understanding of the merits of EBP.

In the broader context, this review and similar studies play a pivotal role in shedding light on the persistent gap between intention and implementation in EBP. By providing objective insights into the challenges and potential solutions, these findings serve as a foundation for developing targeted strategies. The results of this review study will be instrumental in formulating a comprehensive strategy to create conducive conditions for the widespread adoption of EBP in clinical services and healthcare organisations. Ulti-mately, this strategic approach aims to bridge the existing divide between intention and action, fostering a culture where EBP becomes an integral and effective component of professional nursing practice.

Reviewer 4 Report

Comments and Suggestions for Authors

Dear authors,

Many thanks for submitting your work to the journal. According to my review, your manuscript is both well-written and structured. However, I have made some comments that should be addressed. Additional information can be found in the uploaded file.

I wish you a very happy new year!

Best regards

Author Response

Dear Reviewer,

We sincerely thank you for your thorough analysis of our manuscript and for providing valuable insights and recommendations. Your dedication to enhancing the quality of our work is genuinely appreciated.

We have carefully addressed all the "minor comments" you provided and made the necessary corrections. Additionally, we have incorporated a sentence into the introductory paragraph to align with your suggestions and enhance the overall coherence of the manuscript.

In response to your observation about the heterogeneity of health systems, we engaged in extensive discussions among the authors. Ultimately, we agreed to introduce a dedicated subsection that analyzes this heterogeneity. This addition aims to transparently acknowledge variations in the intensity and complexity of factors across different health systems while emphasizing the similarities in influencing Evidence-Based Practice (EBP). This approach enhances the rigour and completeness of our analysis.

To address your concern about the availability of information, we have included detailed data in supplementary files S4 and S5. These files encompass information on the year, country, sample size, study design, population, outcomes, key findings, and more, ensuring transparency and accessibility.

Furthermore, a final paragraph has been appended to section 4.1. of the manuscript, precisely addressing your comment on proposals for future research. We trust this addition aligns with your expectations and contributes meaningfully to the manuscript.

Once again, we express our sincere appreciation for your valuable contribution to our article. Your insights have been instrumental in refining our work, and we look forward to your continued guidance.

All the best,

The authors.

>>>>>>>>>> proposal for future research <<<<<<<<<<

Finally, it should be noted that one of the lines of research to be considered on this topic could focus on the design and conception of studies comparing the results of nursing practices and interventions informed by EBP compared to conventional practices, for example, those handed down by tradition, highlighting the gains obtained in terms of resolving clinical conditions, quality and safety of care, cost-effectiveness of nursing care, patient and nurse satisfaction. From here, also define a set of indicators capable of objectifying these gains and making them visible.